# Molecular Phylogeny and Global Diversity of the Genus *Haploporus* (Polyporales, Basidiomycota)

**DOI:** 10.3390/jof7020096

**Published:** 2021-01-29

**Authors:** Meng Zhou, Yu-Cheng Dai, Josef Vlasák, Yuan Yuan

**Affiliations:** 1Institute of Microbiology, School of Ecology and Nature Conservation, Beijing Forestry University, Beijing 100083, China; zhoumeng9612@bjfu.edu.cn (M.Z.); yuchengdai@bjfu.edu.cn (Y.-C.D.); 2Beijing Advanced Innovation Center for Tree Breeding by Molecular Design, Beijing Forestry University, Beijing 100083, China; 3Biology Centre, Czech Academy of Sciences, Institute of Plant Mol. Biol., Branišovská 31, CZ-370 05 České Budějovice, Czech Republic; vlasak@umbr.cas.cz

**Keywords:** multi-marker analysis, phylogeny, polyporaceae, taxonomy, wood-rotting fungi

## Abstract

Phylogeny and taxonomy of the genus *Haploporus* were carried out based on a larger number of samples covering a wider geographic range including East Asia, South Asia, Europe, and America, and the species diversity of the genus is updated. Four species, *Haploporus bicolor*, *H. longisporus*, *H. punctatus* and *H. srilankensis*, are described as new species based on morphology and molecular phylogenetic analyses inferred from the internal transcribed spacer (ITS), the large subunit nuclear ribosomal RNA gene (nLSU), and the small subunit mitochondrial rRNA gene (mtSSU). *Haploporus bicolor* is characterized by the distinctly different colors between the pore surface and the tubes, small pores measuring 5–7 per mm, and narrow basidiospores measuring 10.5–11.9 × 4.5–5 µm; *H. longisporus* differs from other species in the genus by its large pores measuring 2–3 per mm, hyphae at dissepiment edge with simple septum, and the long basidiospores (up to 22 µm); *H. punctatus* is distinguished by its cushion-shaped basidiocarps, wide fusiform cystidioles with a simple septum at the tips, the absence of dendrohyphidia and the cylindrical to slightly allantoid basidiospores measuring 9–10.8 × 3.8–5 µm; *H. srilankensis* is characterized by its perennial habit, small pores measuring 4–5 per mm, dextrinoid skeletal hyphae, the presence of cystidioles and dendrohyphidia. An identification key to accepted species of *Haploporus* is provided.

## 1. Introduction

*Haploporus* Bondartsev and Singer was established by Singer based on *H. odorus* (Sommerf.) Bondartsev and Singer [1]. The genus was characterized as having annual to perennial, resupinate to pileate basidiocarps, a dimitic to trimitic hyphal system with clamp connections on the generative hyphae, cyanophilous skeletal hyphae, cylindrical to subglobose, colorless, thick-walled, cyanophilous and ornamented basidiospores, and causing a white rot [1,2,3,4,5]. Kotlaba and Pouzar [2] described a genus *Pachykytospora* Kotl. and Pouzar based on *Polyporus tuberculosus* Fr., and it was treated as a synonym of *Haploporus* by Dai et al. [6]. This conclusion was confirmed by molecular analysis [5], and all species introduced in *Pachykytospora* have been transferred to *Haploporus* [2,5,6].

Recently, two species of *Haploporus*, *H. brasiliensis* Nogueira-Melo and Ryvarden and *H. pileatus* Ryvarden, were described from Brazil based on morphology only [7], and four species, *H. angustisporus* Meng Zhou and Y.C. Dai, *H. crassus* Meng Zhou and Y.C. Dai, *H. gilbertsonii* Meng Zhou, Vlasák and Y.C. Dai and *H. microspores* L.L. Shen, Y.C. Dai and B.K. Cui, were introduced as new based on morphological characteristics and molecular phylogenetic analyses [8,9]. So far, 19 species have been accepted in *Haploporus* [2,3,4,5,7,9,10,11].

The aim of the study is to explore the phylogeny and species diversity of *Haploporus* based on samples from Asia, Europe and America. The taxonomy and phylogeny are updated, and four new species, respectively, from China, Ecuador, and Sri Lanka, were confirmed to be members of the *Haploporus*. In this paper, we describe and illustrate those new species.

## 2. Materials and Methods

### 2.1. Morphological Studies

Specimens examined were deposited in the herbarium of the Institute of Microbiology, Beijing Forestry University (BJFC) and the private herbarium of Josef Vlasák (JV) which will forward specimens to the National Museum of Czech Republic in Prague (PRM). Macro-morphological descriptions were based on field notes. Sections of basidiocarps were studied microscopically according to Zhou and Cui [12] at magnifications 1000× using a Nikon Eclipse 80i microscope with phase contrast illumination. Drawings were made with the aid of a drawing tube. Microscopic features, measurements, and drawings were made from sections stained with Cotton Blue and Melzer’s reagent. Basidiospores were measured from sections cut from the tubes. To present basidiospores size variation, the 5% of measurements excluded from each end of the range are given in parentheses. Basidiospore spine lengths were not included in the measurements. Abbreviations include IKI = Melzer’s reagent, IKI+ = amyloid, KOH = 5% potassium hydroxide, CB = Cotton Blue, CB+ = cyanophilous, L = mean spore length (arithmetic average of all spores), W = mean basidiospores width (arithmetic average of all basidiospores), Q = the L/W ratio, and n = number of basidiospores measured from given number of specimens. Special color terms follow Petersen [13]. Herbarium abbreviations follow Thiers [14].

### 2.2. DNA Extraction and Sequencing

A cetyl trimethylammonium bromide (CTAB)rapid plant genome extraction kit (Aidlab Biotechnologies, Beijing, China) was used to obtain PCR products from dried specimens, according to the manufacturer’s instructions with some modifications [15,16]. The internal transcribed spacer (ITS) region was amplified with primer pair ITS5 (GGA AGT AAA AGT CGT AAC AAG G) and ITS4 (TCC TCC GCT TAT TGATAT GC) [17]. The large subunit nuclear ribosomal RNA gene (nLSU) region was amplified with primer pair LR0R (ACC CGC TGA ACT TAA GC) and LR7 (TAC TAC CAC CAA GAT CT) (http://www.biology.duke.edu/fungi/mycolab/primers.htm). The small subunit mitochondrial rRNA gene (mtSSU) region was amplified with primer pair MS1 (CAG CAG TCA AGA ATA TTA GTC AAT G) and MS2 (GCG GAT TAT CGA ATT AAA TAA C) [17]. The PCR procedure for ITS and mtSSU was as follows: initial denaturation at 95 °C for 3 min, followed by 34 cycles at 94 °C for 40 s, 54 °C for ITS and 55 °C for mtSSU for 45 s and 72 °C for 1 min, and a final extension of 72 °C for 10 min. The PCR procedure for nLSU was as follows: initial denaturation at 94 °C for 1 min, followed by 34 cycles at 94 °C for 30 s, 50 °C for 1 min and 72 °C for 1.5 min, and a final extension of 72 °C for 10 min [5]. The PCR products were purified with a Gel Extraction and PCR Purification Combo Kit (Spin-column) in Beijing Genomics Institute, Beijing, P.R. China. The purified products were then sequenced on an ABI-3730-XL DNA Analyzer (Applied Biosystems, Foster City, CA, USA) using the same primers as in the original PCR amplifications.

### 2.3. Phylogenetic Analysis

The sequences generated in this study were aligned with additional sequences downloaded from GenBank (Table 1) using ClustalX [18] and manually adjusted in BioEdit [19]. The sequence quality was checked following Nilsson et al. [20]. *Perenniporia hainaniana* B.K. Cui and C.L. Zhao and *P. medulla-panis* (Jacq.) Donk were used as outgroups, following Shen et al. [5]. Prior to phylogenetic analysis, ambiguous sequences at the start and the end were deleted and gaps were manually adjusted to optimize the alignment. Sequence alignment was deposited at TreeBase (http://purl.org/phylo/treebase; submission ID 27556).

Maximum parsimony (MP), Maximum likelihood (ML) and Bayesian inference (BI) were employed to perform phylogenetic analysis of the three aligned datasets. The three phylogenetic analysis algorithms generated nearly identical topologies for each dataset, thus only the topology from the MP analysis is presented along with statistical values from the MP, ML and BI algorithms. Most parsimonious phylogenies were inferred from the ITS + nLSU + mtSSU, and their combinability was evaluated with the incongruence length difference (ILD) test [23] implemented in PAUP* 4.0b10 [24], under a heuristic search and 1000 homogeneity replicates giving a P value of 1.000, much greater than 0.01, which means there is no discrepancy among the two loci in reconstructing phylogenetic trees. Phylogenetic analysis approaches followed [22]. The tree construction procedure was performed in PAUP* version 4.0b10 [24]. All characters were equally weighted, and gaps were treated as missing data. Trees were inferred using the heuristic search option with TBR branch swapping and 1000 random sequence additions. Max-trees were set to 5000, branches of zero length were collapsed and all parsimonious trees were saved. Clade robustness was assessed using a bootstrap (BT) analysis with 1000 replicates [25]. Descriptive tree statistics tree length (TL), consistency index (CI), retention index (RI), rescaled consistency index (RC), and homoplasy index (HI) were calculated for each maximum parsimonious tree (MPT) generated.

jModeltest v.2.17 [26] was used to determine the best-fit evolution model of the combined dataset for Maximum likelihood (ML) and Bayesian inference (BI). Four unique partitions were established, GTR + I + G was the selected substitution model for each partition. RaxmlGUI 1.2 [27,28] was used for ML analysis. All parameters in the ML analysis used default settings. Statistical support values were obtained using non-parametric bootstrapping with 1000 replicates. The Bayesian inference (BI) was conducted with MrBayes 3.2.6 [29] in two independent runs, each of which had four chains for 10 million generations and started from random trees. Trees were sampled every 1000th generation. The first 25% of sampled trees were discarded as burn-in, whereas other trees were used to construct a 50% majority consensus tree and for calculating Bayesian posterior probabilities (BPPs).

Phylogenetic trees were visualized using Treeview [30]. Branches that received bootstrap support for Maximum likelihood (BS), Maximum parsimony (BP) and Bayesian posterior probabilities (BPP) greater than or equal to 75% (BS/BP) and 0.95 (BPP) were considered as significantly supported, respectively.

## 3. Results

### 3.1. Molecular Phylogeny

The ITS-based phylogeny included ITS sequences from 46 fungal collections representing 23 species. The dataset had an aligned length of 720 characters, of which 333 characters are constant, 63 are variable and parsimony-uninformative, and 324 are parsimony-informative. MP analysis yielded a tree (TL = 945, CI = 0. 624, RI = 0.872, RC = 0.544, HI = 0.376). The best model for the ITS sequences dataset estimated and applied in the BI was GTR+I+G. BI resulted in a similar topology with an average standard deviation of split frequencies = 0.006018 to MP analysis, and thus only the MP tree is provided. Both BT values (≥50%) and BPPs (≥0.90) are shown at the nodes (Figure 1).

The combined three gene (ITS + nLSU + mtSSU) sequences dataset from 46 fungal specimens representing 23 taxa did not show any conflicts in tree topology for the reciprocal bootstrap trees, which allowed us to combine them (*p* > 0.01). The dataset had an aligned length of 2706 characters, of which 1792 characters are constant, 220 are variable and parsimony-uninformative, and 694 are parsimony-informative. Maximum parsimony analysis yielded a tree (TL = 1929, CI = 0.627, RI = 0.847, RC = 0.531, HI = 0.373). The best model for the combined ITS + nLSU + mtSSU dataset estimated and applied in the Bayesian analysis was GTR + I + G. Bayesian analysis resulted in a similar topology with an average standard deviation of split frequencies = 0.005181 to MP analysis, and thus only the MP tree is provided. Both BT values (≥50%) and BPPs (≥0.90) are shown at the nodes (Figure 2).

In both ITS + nLSU + mtSSU- and ITS-based phylogenies (Figure 1 and Figure 2), four new well-supported lineages were identified. Among them are two well-supported terminal clades and two isolated branches. Four specimens from Sri Lanka formed a well-supported clade (98% MP 100% ML and 1.00 BI), named as Haploporus srilankensis, sister to H. angustisporus. And two specimens of Haploporus punctatus nested in the same clade as H. srilankensis and H. angustisporus. In addition, samples of Haploporus bicolor and H. longisporus formed two distinct lineages, H. bicolor is sister to H. septatus L.L. Shen, Y.C. Dai and B.K. Cui; but H. longisporus is an independent lineage.

### 3.2. Taxonomy

***Haploporus bicolor*** Y.C. Dai, Meng Zhou and Yuan Yuan, sp. Nov. Figure 3 and Figure 4

MycoBank: MB838450

**Diagnosis**—Differs from other *Haploporus* species by the distinct different colors between the pore surface and the tubes, no-dextrinoid skeletal hyphae, small pores measuring 5–7 per mm, the presence of fusiform cystidioles and abundant dendrohyphidia, and narrow basidiospores measuring 10.5–11.9 × 4.5–5 µm.

**Type**—China, Yunnan, Wenshan, Xichou County, Xiaoqiaogou Forest Farm, E 104°41′, N 23°21′, on a fallen angiosperm branch, 29 June 2019, Yu-Cheng Dai, Dai 19951 (Holotype BJFC 031625).

**Etymology—*Bicolor*** (Lat.): referring to the species having different colors between the pore surface and the tubes.

**Fruiting body**—Basidiocarp annual, resupinate, difficult to separate from the substrate, soft corky and white to cream when fresh, become buff when bruised, corky when dry, up to 5 cm long, 2 cm wide and 0.7 mm thick at the center. Pore surface cream to buff with peach tint when dry; sterile margin distinct, white, up to 1 mm; pores round to angular, 5–7 per mm; dissepiments thick, entire. Subiculum clay buff, corky, up to 0.4 mm thick. Tubes clay buff, corky, up to 0.3 mm long.

**Hyphal structure**—Hyphal system dimitic; generative hyphae bearing clamp connections, hyaline, thin-walled; skeletal hyphae dominant, thick-walled, frequently branched, IKI–, CB+; tissues unchanging in KOH.

**Subiculum**—Generative hyphae inconspicuous, hyaline, thin-walled, 1–2.5 µm in diameter (diam); skeletal hyphae dominant, distinctly thick-walled with a narrow lumen, frequently branched, flexuous, interwoven, 1–2.5 µm in diam.

**Tubes**—Generative hyphae infrequent, hyaline, thin-walled, 1–1.5 µm in diam; skeletal hyphae dominant, distinctly thick-walled with a narrow lumen, frequently branched, flexuous, interwoven, 1–2 µm in diam. Cystidia absent; cystidioles present, fusiform to ventricose, hyaline, thin-walled, 13.5–22 × 4–6.5 µm. Basidia barrel-shaped to pear-shaped with 4-sterigmata and a basal clamp connection, 14–25 × 6–11.5 µm; basidioles dominant, pear-shaped, slightly smaller than basidia. Dendrohyphidia abundant, hyaline, thin-walled. Some irregular-shaped crystals present among tube trama.

**Spores**—Basidiospores oblong ellipsoid to subcylindrical, hyaline, thick-walled with tuberculate ornamentation, IKI–, CB+, (10–)10.5–11.9(–12) × (4.1–)4.5–5(–5.1) µm, L = 11.10 µm, W = 4.86 µm, Q = 2.28 (n = 30/1).

***Haploporus longisporus*** Y.C. Dai, Meng Zhou and Vlasák, sp. Nov. Figure 5 and Figure 6

MycoBank: MB838451

**Diagnosis**—Differs from other *Haploporus* species by the non-dextrinoid skeletal hyphae, large pores measuring 2–3 per mm, hyphae at dissepiment edge with simple septum, and the long cylindrical basidiospores measuring 18.2–22 × 7–9 µm.

**Type**—Ecuador, Pichincha, Vicodin svah volcan Pasochoa, W 78°29′, S 0°25′, on a dead angiosperm branch, June 2019, Josef Vlasák, JV1906/C11-J (Holotype PRM, isotypes BJFC 032989 and JV).

**Etymology—*Longisporus*** (Lat.): referring to the species having the distinct long basidiospores.

**Fruiting body**—Basidiocarp annual, resupinate, difficult to separate from the substrate, corky when dry, up to 10 cm long, 1.5 cm wide and 2 mm thick at the center. Pore surface cream to pale buff when dry; sterile margin indistinct; pores round to angular, 2–3 per mm; dissepiments thick, entire. Subiculum olivaceous buff, corky, up to 1 mm thick. Tubes olivaceous buff, corky, 1 mm long.

**Hyphal structure**—Hyphal system dimitic; generative hyphae bearing clamp connections, hyaline, thin-walled; skeletal hyphae dominant, thick-walled, frequently branched, IKI–, CB+; tissues unchanging in KOH.

**Subiculum**—Generative hyphae infrequent, hyaline, thin-walled, occasionally branched, 2–3 µm in diam; skeletal hyphae dominant, distinctly thick-walled with a narrow to wide lumen, frequently branched, flexuous, interwoven, 2.5–5 µm in diam.

**Tubes**—Generative hyphae hyaline, thin-walled, frequently branched, 1.5–3 µm in diam; skeletal hyphae dominant, distinctly thick-walled with a narrow to medium lumen, frequently branched, flexuous, interwoven, 2–5 µm in diam. Cystidia absent; cystidioles present, fusiform to slim clavate, hyaline, thin-walled, 20–33 × 3.5–5 µm. Basidia more or less barrel-shaped, sometimes constricted at middle, with 4-sterigmata and a basal clamp connection, 37–50 × 12–16 µm; basidioles dominant, clavate to pear-shaped, slightly smaller than basidia, sometimes with a large guttule. Hyphae at dissepiment edge usually thick-walled with a simple septum. Dendrohyphidia present, hyaline, thin-walled. Some irregular-shaped crystals present among tube trama.

**Spores**—Basidiospores cylindrical, hyaline, thick-walled with tuberculate ornamentation, sometimes with a few guttules, IKI–, CB+, (18–)18.2–22(–22.5) × (6.5–)7–9(–10) µm, L = 20.6 µm, W = 7.86 µm, Q = 2.62 (*n* = 30/1).

***Haploporus punctatus*** Y.C. Dai, Meng Zhou and Yuan Yuan, sp. Nov. Figure 7 and Figure 8

MycoBank: MB 838452

**Diagnosis**—Differs from other *Haploporus* species by its cushion-shaped basidiocarps, dextrinoid skeletal hyphae and basidiospores, wide fusiform cystidioles with a simple septum at the tips, the absence of dendrohyphidia and the cylindrical to slightly allantoid basidiospores measuring 9–10.8 × 3.8–5 µm.

**Type**—Sri Lanka, Wadduwa, South Blogoda Lake, E 79°57′, N 6°41′, on a dead angiosperm branch, 28 Feburary 2019, Yu-Cheng Dai, Dai 19525 (Holotype BJFC 031204).

**Etymology**—***Punctatus*** (Lat.): referring to the species having cushion-shaped basidiocarps.

**Fruiting body**—Basidiocarps annual, resupinate, difficult to separate from the substrate, corky when dry, cushion-shaped, distinctly thickened in center and receding at margin, up to 3 cm long, 3 cm wide and 4 mm thick at center. Pore surface pale buff when dry; sterile margin distinct, cream, up to 2 mm; pores round to angular, 3–5 per mm; dissepiments thick, entire. Subiculum olivaceous buff to clay buff, corky, up to 0.5 mm thick. Tubes buff, corky, up to 3.5 mm long.

**Hyphal structure**—Hyphal system dimitic; generative hyphae bearing clamp connections, hyaline, thin-to-slightly-thick-walled; skeletal hyphae dominant, thick-walled, frequently branched, dextrinoid, CB+; tissues unchanging in KOH.

**Subiculum**—Generative hyphae frequent, hyaline, thin-to-slightly-thick-walled, frequently branched, 1.5–2.5 µm in diam; skeletal hyphae dominant, hyaline, distinctly thick-walled with a narrow lumen to subsolid, frequently branched, flexuous, interwoven, 1.5–3 µm in diam.

**Tubes**—Generative hyphae frequent, hyaline, thin-walled, frequently branched, 1–2 µm in diam; skeletal hyphae dominant, distinctly thick-walled with a narrow lumen to subsolid, frequently branched, flexuous, interwoven, 1–2 µm in diam. Cystidia absent; cystidioles present, hyaline, thin-walled, wide fusiform, sometimes with a simple septum at the tips, 13–28 × 4–9 µm. Basidia barrel-shaped with 4-sterigmata and a basal clamp connection, occasionally with a large guttule, 16–28 × 6–10 µm; basidioles dominant, capitate to pear-shaped, distinctly smaller than basidia. Dendrohyphidia absent. Irregular-shaped crystals present abundantly among tube trama.

**Spores**—Basidiospores cylindrical to slightly allantoid, hyaline, thick-walled with tuberculate ornamentation, sometimes with a few guttules, dextrinoid, CB+, 9–10.8(–11) × 3.8–5(–5.3) µm, L = 10.36 µm, W = 4.36 µm, Q = 2.37–2.52 (*n* = 60/2).

**Additional specimen examined (paratype)**—Sri Lanka, Avissawella, Salgala Forest, E 80°16′, N 7°5′, on a fallen dead angiosperm branch, 28 February 2019, Dai 19628 (BJFC 031305).

***Haploporus srilankensis*** Y.C. Dai, Meng Zhou and Yuan Yuan, sp. nov. Figure 9 and Figure 10

MycoBank: MB838453

**Diagnosis**—Differs from other *Haploporus* species by the combination of perennial habit, small pores measuring 4–5 per mm, dextrinoid skeletal hyphae, the presence of cystidioles and dendrohyphidia, and short basidiospores measuring 8.5–11.5 × 4–5.2 µm.

**Type**—Sri Lanka, Wadduwa, South Blogoda Lake, E 79°57′, N 6°41′, on a dead branch of a living angiosperm tree, 28 February 2019, Yu-Cheng Dai, Dai 19523 (Holotype BJFC 031202).

**Etymology**—***Srilankensis*** (Lat.): referring to the species occurrence in Sri Lanka.

**Fruiting body**—Basidiocarps perennial, resupinate, difficult to separate from the substrate, corky when fresh and dry, up to 14 cm long, 4 cm wide and 2.5 mm thick at the center. Pore surface pale buff when fresh and pinkish buff to salmon when dry; sterile margin distinct, white to cream, up to 2 mm; pores round, 4–5 per mm; dissepiments thick, entire. Subiculum buff-yellow, corky, up to 0.5 mm thick. Tubes buff, corky, up to 2 mm long.

**Hyphal structure**—Hyphal system dimitic; generative hyphae bearing clamp connections, hyaline, thin-walled; skeletal hyphae dominant, thick-walled with a narrow lumen, frequently branched, dextrinoid, CB+; tissues unchanging in KOH.

**Subiculum**—Generative hyphae infrequent, hyaline, thin-walled, frequently branched, 1.5–2 µm in diam; skeletal hyphae dominant, hyaline, distinctly thick-walled with a narrow lumen, frequently branched, flexuous, interwoven, 1.5–2 µm in diam.

**Tubes**—Generative hyphae hyaline, thin-walled, frequently branched, 1–2 µm in diam; skeletal hyphae dominant, distinctly thick-walled with a narrow lumen, frequently branched, flexuous, interwoven, 1–2.5 µm in diam. Cystidia absent; cystidioles present, distinctly fusiform, hyaline, thin-walled, 17–24 × 4–6 µm. Basidia clavate to ampullaceous with 4-sterigmata and a basal clamp connection, occasionally with a few small guttules, 18–28 × 5–9 µm; basidioles dominant, capitate to pear-shaped, distinctly wider than basidia. Dendrohyphidia present, hyaline, thin-walled. Some irregular-shaped crystals present among tube trama.

**Spores**—Basidiospores oblong ellipsoid, hyaline, thick-walled, with tuberculate ornamentation, occasionally with a large guttule, dextrinoid, CB+, (8.3–)8.5–11(–12) × (3.9–)4–5.2(–5.5) µm, L = 9.80 µm, W = 4.56 µm, Q = 2.07–2.21 (*n* = 90/3).

**Additional specimens examined (paratypes)**—Sri Lanka, Wadduwa, South Blogoda Lake, E 79°57′, N 6°41′, on a dead angiosperm branch, 28 Feb. 2019, Dai 19524 (BJFC 031203), Dai 19526 (BJFC 031205), Dai 19530 (BJFC 031209), Dai 19534 (BJFC 031213).

## 4. Discussion

In both ITS and ITS + nLSU + mtSSU-based phylogenies (Figure 1 and Figure 2), *Haploporus bicolor* is sister to *H. septatus*. Morphologically, *Haploporus bicolor* resembles *H. septatus* by sharing resupinate basidiocarps with approximately the same-sized small pores (5–7 per mm), but *H. septatus* differs from *H. bicolor* by wider basidiospores (8.5–11 × 5–6 µm vs. 10.5–11.9 × 4.5–5 µm), dextrinoid skeletal hyphae and absence of dendrohyphidia [5]. In addition, *Haploporus bicolor* is also related to *H. crassus*, *H. papyraceus* (Cooke) Y.C. Dai and Niemelä, *H. subpapyraceus* L.L. Shen, Y.C. Dai and B.K. Cui in our phylogenies (Figure 1 and Figure 2). However, these three species have larger pores (3–5 per mm) and wider basidiospores (width > 5 µm), in which they obviously differ from *Haploporus bicolor* [5,9,31].

Phylogenetically, *Haploporus longisporus* is an independent lineage. Morphologically, *Haploporus longisporus*, *H. crassus*, *H. pirongia* (G. Cunn.) Meng Zhou, Y.C. Dai and T.W. May and *H. septatus* share similar hyphae at dissepiment edge (thick-walled with a simple septum). However, *Haploporus crassus* is distinguished from *H. longisporus* by its thick-walled basidia and shorter basidiospores (13.5–16.5 µm vs. 18.2–22 µm) [9]. *Haploporus pirongia* differs from *H. longisporus* in having smaller basidiospores (11–14 × 5.2–7 µm vs. 18.2–22 × 7–9 µm) [9]. *Haploporus septatus* differs from *H. longisporus* by the dextrinoid skeletal hyphae and smaller pores and basidiospores (5–6 per mm, 8.5–11 × 5–6 μm) [5].

In both ITS- and ITS + nLSU + mtSSU-based phylogenies (Figure 1 and Figure 2), the studied Sri Lankan samples formed two independent lineages, and they are represented by two new species: *Haploporus punctatus* and *H. srilankensis*. Both new species are closely related to *Haploporus angustisporus* and these three species are nested in a clade (Figure 1 and Figure 2). Morphologically, *Haploporus angustisporus* differs from *H. punctatus* by having longer basidiospores (10–13.5 × 4–5 µm vs. 9–10.8 × 3.8–5 µm) and obviously narrow fusiform cystidioles without simple septum at the tips [9]. *Haploporus angustisporus* differs from *H. srilankensis* by its annual habit, the absence of dendrohyphidia, relatively longer basidiospores (10–13.5 × 4–5 µm vs. 8.5–11.5 × 4–5 µm) [9]. Although both *Haploporus punctatus* and *H. srilankensis* occur in Sri Lanka, there are 17 base pairs differences between *H. punctatus* and *H. srilankensis*, which amounts to > 2% nucleotide differences in the ITS regions. In addition, *Haploporus srilankensis* differs from *H. punctatus* by its perennial habit, the presence of dendrohyphidia and cystidioles without septa.

*Haploporus punctatus* may be confused with *H. subpapyraceus* and *H. crassus* in having approximately the same pores size (3–5 per mm) and cystidioles with a simple septum at the tips. However, *Haploporus subpapyraceus* has wider basidiospores (9–12 × 5.5–8 μm vs. 9–10.8 × 3.8–5 µm) [5]. *Haploporus crassus* differs from *H. punctatus* by its longer basidiospores (13.5–16.5 × 4–5 µm vs. 9–10.8 × 3.8–5 µm) and the unique thick-walled basidia [9].

All the species of *Haploporus* with molecular evidence and close to our new species are discussed above. Furthermore, two species of *Haploporus*, *H. brasiliensis* and *H. pileatus*, were described from Brazil based on morphology only [7]. *Haploporus brasiliensis* distinctly differs from our four new species by its obviously smaller basidiospores and the absence of cystidioles and dendrohyphidia [7]. *Haploporus pileatus* is distinguished from all the species in this genus by its distinct pileate basidiocarps [7].

Currently, 23 species are accepted in *Haploporus*, and their main morphological characteristics are listed in Table 2. A key to accepted species of *Haploporus* is provided as follows.

## Figures and Tables

**Figure 1 jof-07-00096-f001:**
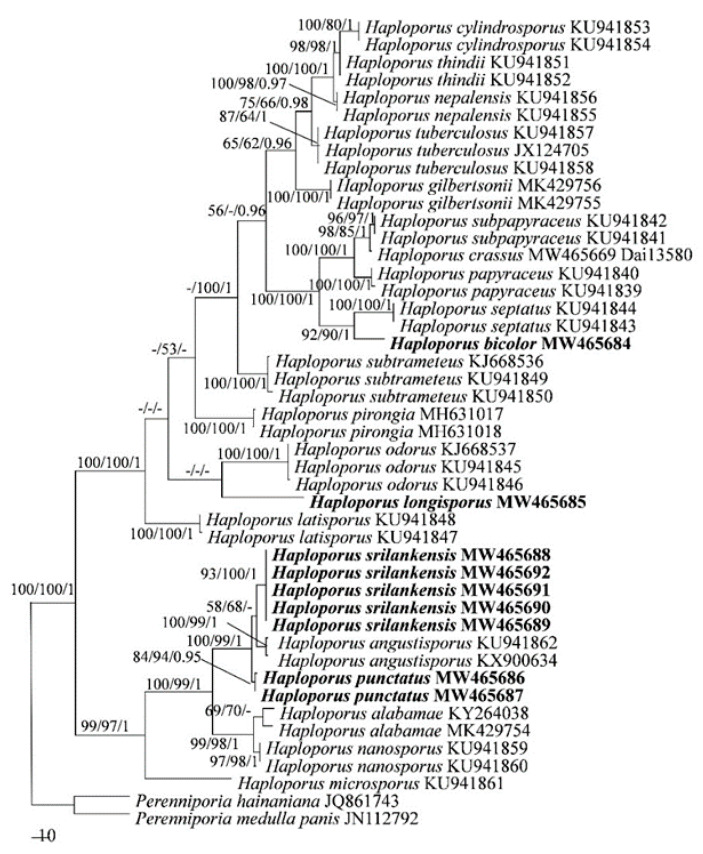
Maximum parsimony strict consensus tree illustrating the phylogeny of *Haploporus* based on internal transcribed spacer (ITS) sequences. Branches are labeled with parsimony bootstrap values ≥ 50% and Bayesian posterior probabilities ≥ 0.90.

**Figure 2 jof-07-00096-f002:**
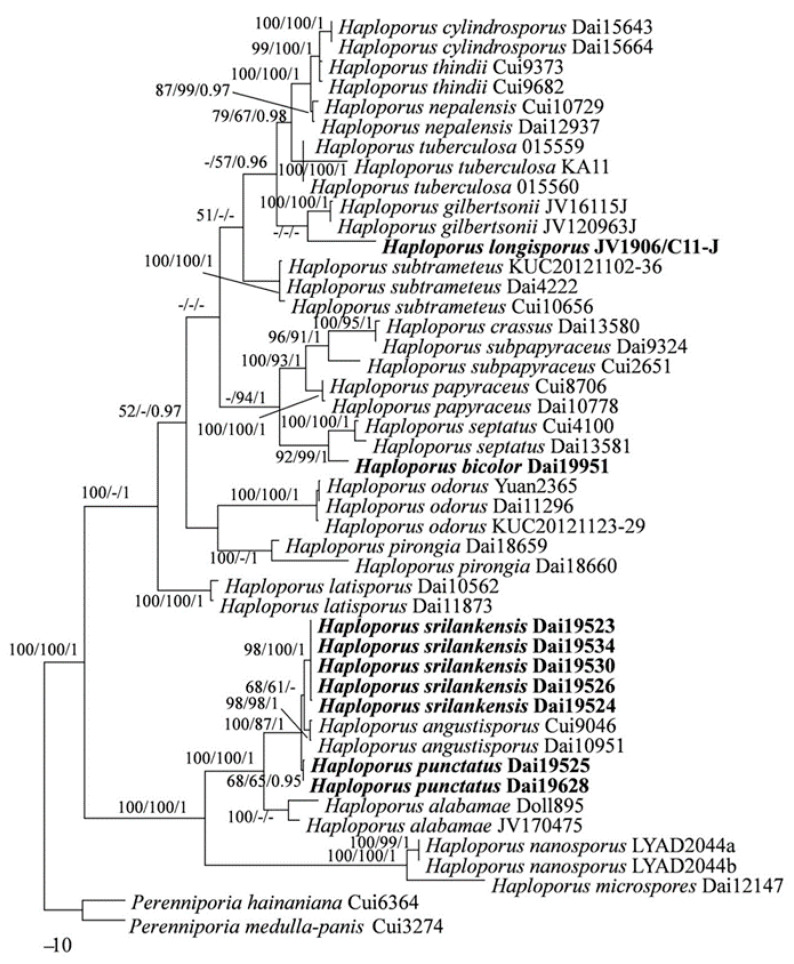
Maximum parsimony strict consensus tree illustrating the phylogeny of *Haploporus* based on ITS + nLSU + mtSSU sequences. Branches are labeled with parsimony bootstrap values ≥ 50% and Bayesian posterior probabilities ≥ 0.90.

**Figure 3 jof-07-00096-f003:**
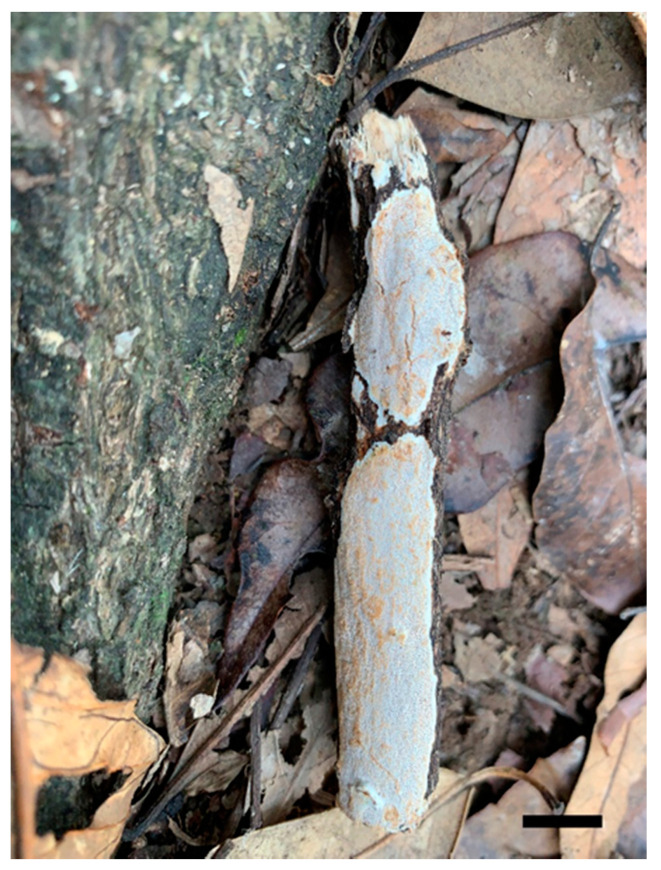
Basidiocarps of *Haploporus bicolor* (Holotype). Scale bar = 1.0 cm.

**Figure 4 jof-07-00096-f004:**
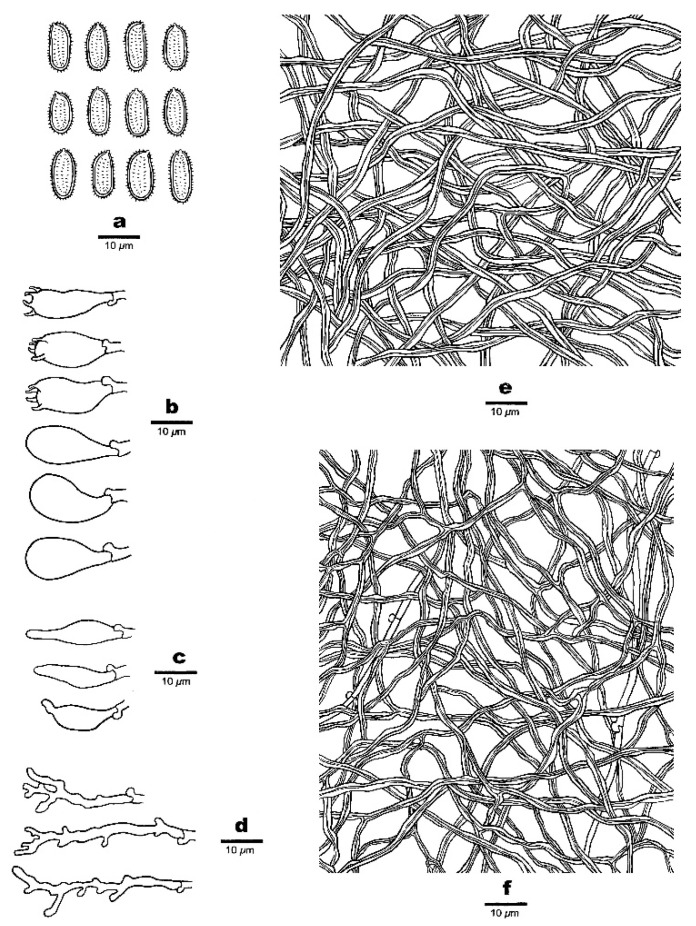
Microscopic structures of *Haploporus bicolor* (Holotype). (**a**). Basidiospores. (**b**). Basidia and basidioles. (**c**). Cystidioles. (**d**). Dendrohyphidia. (**e**). Hyphae from subiculum. (**f**). Hyphae from trama.

**Figure 5 jof-07-00096-f005:**
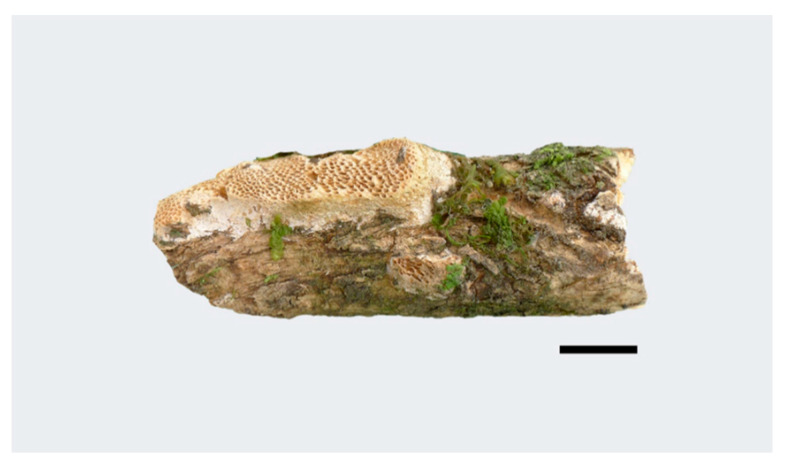
Basidiocarp of *Haploporus longisporus* (Holotype). Scale bar = 1.0 cm.

**Figure 6 jof-07-00096-f006:**
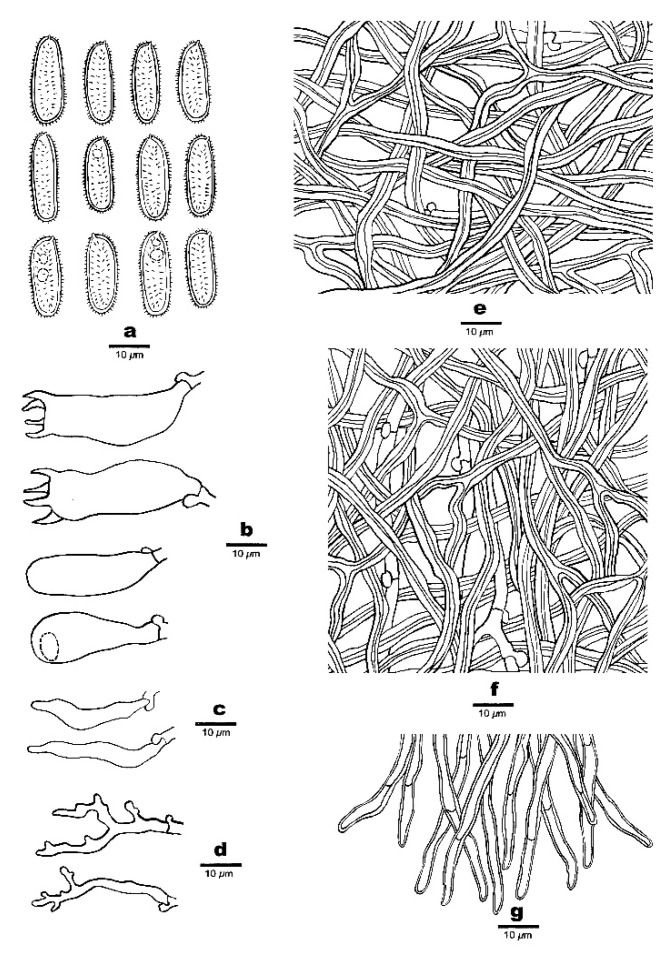
Microscopic structures of *Haploporus longisporus* (Holotype). (**a**). Basidiospores. (**b**). Basidia and basidioles. (**c**). Cystidioles. (**d**). Dendrohyphidia. (**e**). Hyphae from subiculum. (**f**). Hyphae from trama. (**g**). Hyphae at dissepiment edge.

**Figure 7 jof-07-00096-f007:**
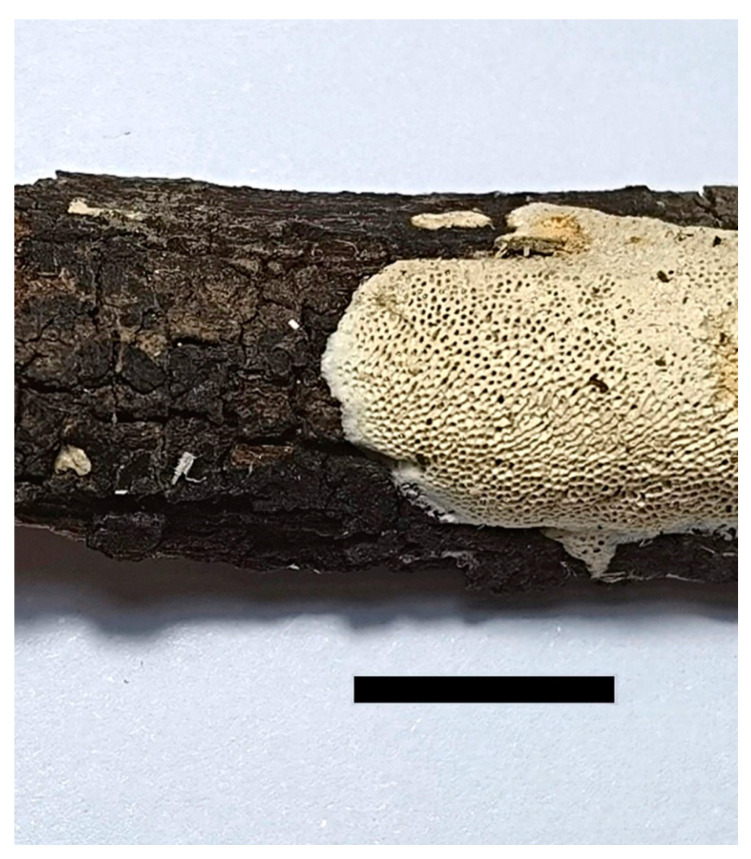
Basidiocarp of *Haploporus punctatus* (Holotype). Scale bar = 1.0 cm.

**Figure 8 jof-07-00096-f008:**
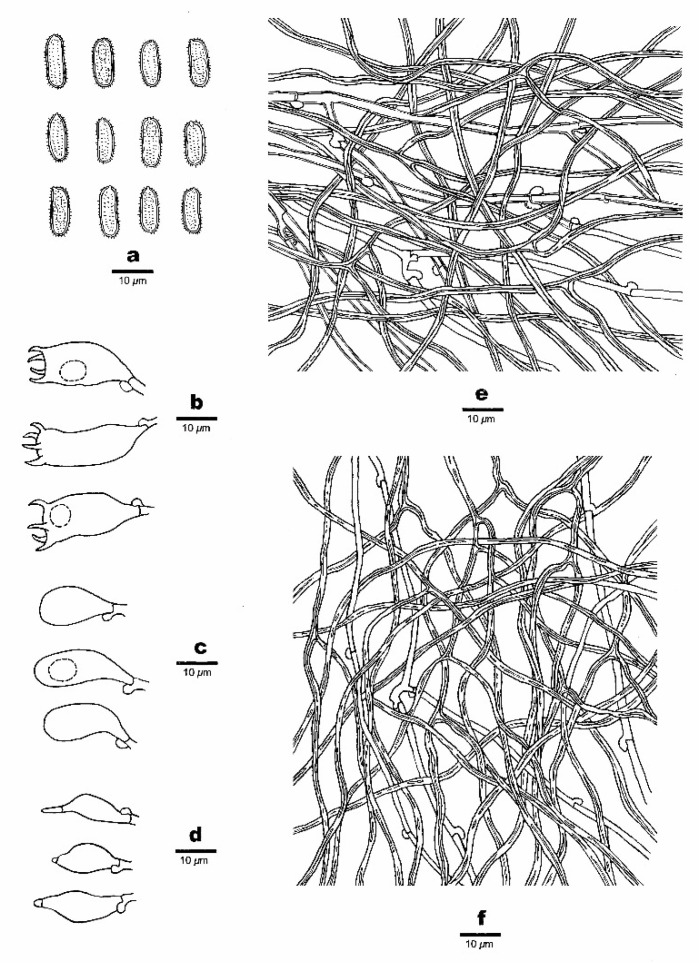
Microscopic structures of *Haploporus punctatus* (Holotype). (**a**). Basidiospores. (**b**). Basidia. (**c**). Basidioles. (**d**). Cystidioles. (**e**). Hyphae from subiculum. (**f**). Hyphae from trama.

**Figure 9 jof-07-00096-f009:**
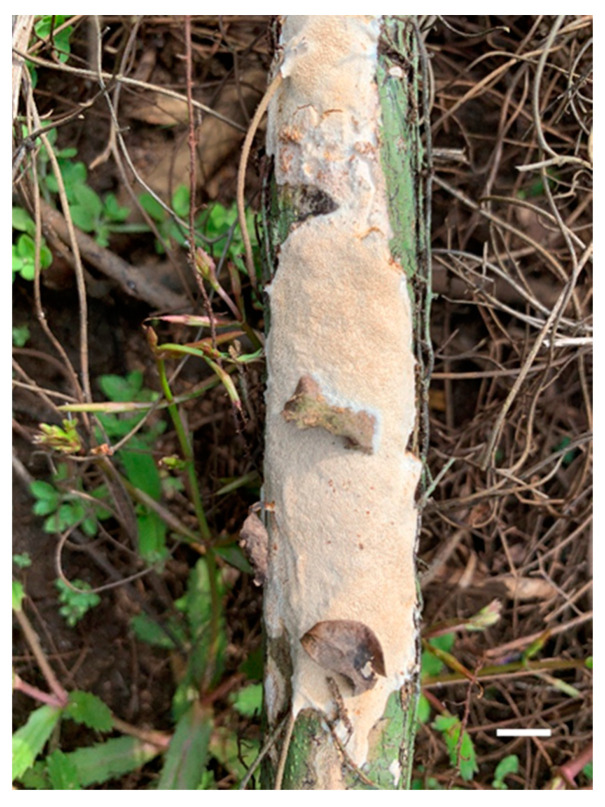
Basidiocarp of *Haploporus srilankensis* (Holotype). Scale bar = 1.0 cm.

**Figure 10 jof-07-00096-f010:**
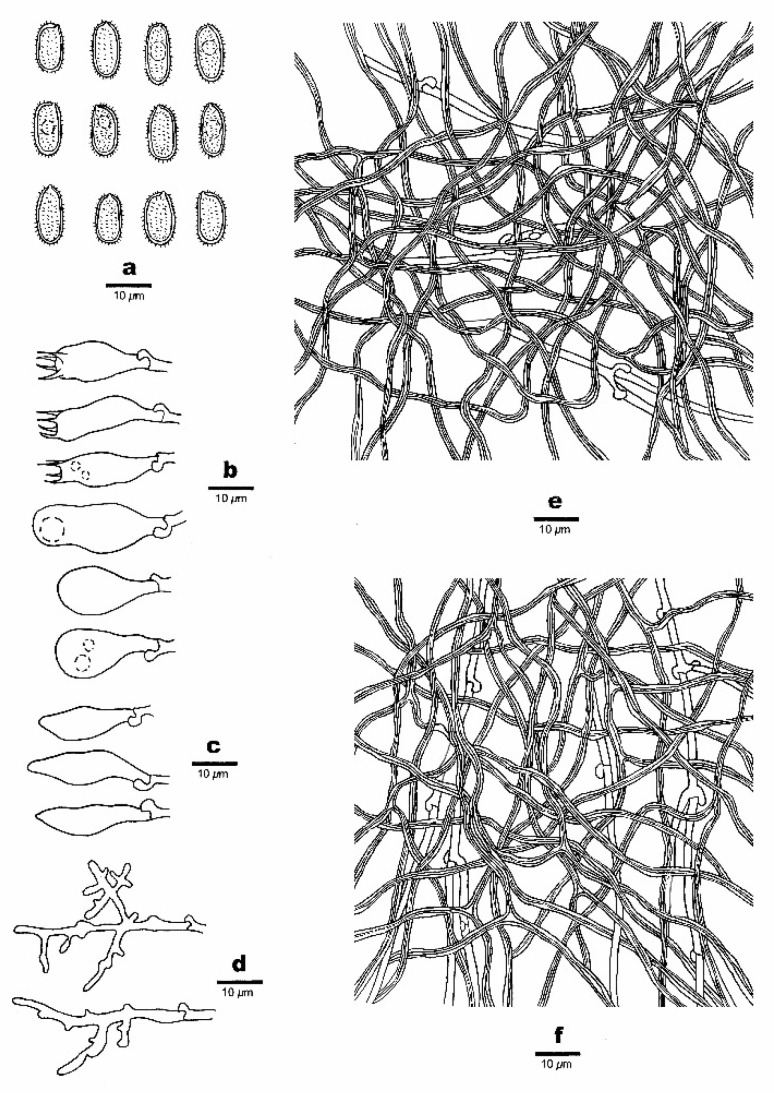
Microscopic structures of *Haploporus srilankensis* (Holotype). (**a**). Basidiospores. (**b**). Basidia and basidioles. (**c**). Cystidioles. (**d**). Dendrohyphidia. (**e**). Hyphae from subiculum. (**f**). Hyphae from trama.

**Table 1 jof-07-00096-t001:** Information on the sequences used in this study.

Species	Sample No.	Location	GenBank Accession No.			Reference
			ITS	nLSU	mt-SSU	
*Haploporus* *alabamae*	Dollinger 895	USA	KY264038	MK433606	MW463004	[9]
*H.* *alabamae*	JV 1704/75	Costa Rica	MK429754	MK433607	MW463005	[9]
*H. angustisporus*	Cui 9046	China	KU941862	KU941887		[9]
*H. angustisporus*	Dai 10951	China	KX900634	KX900681	MW463006	[9]
***H. bicolor***	**Dai19951**	**China**	**MW465684**	**MW462995**		**Present study**
*H. crassus*	Dai 13580	China	MW465669	KU941886		[9]
*H. cylindrosporus*	Dai 15643	China	KU941853	KU941877	KU941902	[5]
*H. cylindrosporus*	Dai 15664	China	KU941854	KU941878	KU941903	[5]
*H.* *gilbertsonii*	JV 1209/63-J	USA	MK429755	MK433608		[9]
*H.* *gilbertsonii*	JV 1611/5-J	USA	MK429756	MK433609	MW463007	[9]
*H. latisporus*	Dai 11873	China	KU941847	KU941871	MW463008	[5]
*H. latisporus*	Dai 10562	China	KU941848	KU941872	KU941897	[5]
***H. longisporus***	**JV1906/C11-J**	**Ecuador**	**MW465685**	**MW462996**		**Present study**
*H. microsporus*	Dai 12147	China	KU941861	KU941885		[5]
*H. nanosporus*	LYAD 2044°	Gabon	KU941859	KU941883	KU941908	[5]
*H. nanosporus*	LYAD 2044b	Gabon	KU941860	KU941884		[5]
*H. nepalensis*	Dai 12937	China	KU941855	KU941879	KU941904	[5]
*H. nepalensis*	Cui 10729	China	KU941856	KU941880	KU941905	[5]
*H. odorus*	Dai 11296	China	KU941845	KU941869	KU941894	[5]
*H. odorus*	Yuan 2365	China	KU941846	KU941870	KU941895	[5]
*H. odorus*	KUC20121123-29	Republic of Korea	KJ668537	KJ668390		
*H. papyraceus*	Dai 10778	China	KU941839	KU941863	KU941888	[5]
*H. papyraceus*	Cui 8706	China	KU941840	KU941864	KU941889	[5]
*H. pirongia*	Dai 18659	Australia	MH631017	MH631021	MW463009	[9]
*H. pirongia*	Dai 18660	Australia	MH631018	MH631022	MW463010	[9]
***H. punctatus***	**Dai19525**	**Sri Lanka**	**MW465686**	**MW462997**		**Present study**
***H. punctatus***	**Dai19628**	**Sri Lanka**	**MW465687**	**MW462998**	**MW463011**	**Present study**
*H. septatus*	Dai 13581	China	KU941843	KU941867	KU941892	[5]
*H. septatus*	Cui 4100	China	KU941844	KU941868	KU941893	[5]
***H. srilankensis***	**Dai19523**	**Sri Lanka**	**MW465688**	**MW462999**	**MW463012**	**Present study**
***H. srilankensis***	**Dai19524**	**Sri Lanka**	**MW465689**	**MW463000**		**Present study**
***H. srilankensis***	**Dai19526**	**Sri Lanka**	**MW465690**	**MW463001**		**Present study**
***H. srilankensis***	**Dai19530**	**Sri Lanka**	**MW465691**	**MW463002**	**MW463013**	**Present study**
***H. srilankensis***	**Dai19534**	**Sri Lanka**	**MW465692**	**MW463003**	**MW463014**	**Present study**
*H. subpapyraceus*	Dai 9324	China	KU941841	KU941865	KU941910	[5]
*H. subpapyraceus*	Cui 2651	China	KU941842	KU941866	KU941891	[5]
*H. subtrameteus*	Dai 4222	China	KU941849	KU941873	KU941898	[5]
*H. subtrameteus*	Cui 10656	China	KU941850	KU941874	KU941899	[5]
*H. subtrameteus*	KUC20121102-36	Republic of Korea	KJ668536	KJ668389		
*H. thindii*	Cui 9373	China	KU941851	KU941875	KU941900	[5]
*H. thindii*	Cui 9682	China	KU941852	KU941876	KU941901	[5]
*H. tuberculosus*	15559	Sweden	KU941857	KU941881	KU941906	[5]
*H. tuberculosus*	15560	Austria	KU941858	KU941882	KU941907	[5]
*H. tuberculosus*	KA11 (GB)	Sweden	JX124705			[21]
*Perenniporia hainaniana*	Cui 6364	China	JQ861743	JQ861759	KF051044	[16]
*P. medulla-panis*	Cui 3274	China	JN112792	JN112793	KF051043	[22]

New sequences are shown in bold.

**Table 2 jof-07-00096-t002:** Main morphological characters of *Haploporus* species.

Species	Type Locality	Basidiomata	Pore Surface	Pores/mm	Hyphae	Cystidioles	Basidiospores	Dendrohy-Phidia	References
*H. alabamae*	USA	annual, resupinate	cream when fresh, pale brown when dry	3–5	trimitic, skeletal hyphae IKI–	fusiform	ovoid to ellipsoid, 10–12 × 4–6 μm	absent	[2,8]
*H. angustisporus*	China	annual, resupinate	cream to pale yellowish brown when fresh, brownish when bruised, olivaceous buff to pale brown when dry	3–5	dimitic, skeletal hyphae dextrinoid	fusiform	oblong ellipsoid, 10–13.5 × 4–5 µm	absent	[9]
*H. bicolor*	China	annual, resupinate	cream to buff with peach tint when dry	5-7	dimitic, skeletal hyphae IKI–	fusiform to ventricose	oblong ellipsoid to subcylindrical, 10.5–11.9 × 4.5–5 µm	present	Present study
*H. brasiliensis*	Brazil	annual, resupinate	ochraceous buff	1-3	dimitic, skeletal hyphae dextrinoid	absent	oblong ellipsoid, 6–8 × 4–5 µm	absent	[7]
*H. crassus*	China	annual, resupinate	white to cream when fresh, buff-yellow when dry	3–5	dimitic, skeletal hyphae IKI–	ventricose with a simple septum	oblong ellpsoid, 13.5–16.5 × 7.5–9.5 µm	absent	[9]
*H. cylindrosporus*	China	annual, resupinate	white to cream when fresh, pinkish buff to clay-buff when dry	4–5	dimitic, skeletal hyphae dextrinoid	absent	cylindrical, 10–11.5 × 4.5–5 μm	absent	[5]
*H. gilbertsonii*	USA	annual, resupinate	pale buff to buff when dry	2–3	dimitic, skeletal hyphae IKI–	fusiform	oblong ellipsoid, 12–15 × 6–8 µm	absent	[9]
*H. latisporus*	China	annual, resupinate	white to cream when fresh, clay-buff when dry	1–3	dimitic, skeletal hyphae IKI–	fusiform	oblong ellipsoid to ellipsoid, 13–18.5 × 9–10 μm	absent	[3,8]
*H. longisporus*	Ecuador	annual, resupinate	cream to pale buff when dry	2–3	dimitic, skeletal hyphae IKI–	fusiform to clavate	cylindrical, 18.2–22 × 7–9 µm	present	Present study
*H. microsporus*	China	annual, resupinate	pinkish buff to clay-buff when dry	7–9	dimitic, skeletal hyphae dextrinoid	fusiform	ellipsoid, 5.3–6.7 × 3–4.1 µm	present	[8]
*H. nanosporus*	Gabon	annual to perennial, resupinate	cream when fresh, buff to clay-buff when dry	7–8	trimitic, skeletal hyphae dextrinoid	absent	ellipsoid, 5–6 × 3–4 μm	absent	[4]
*H. nepalensis*	Nepal	annual, resupinate	white to cream when fresh, clay-pink to clay-buff when dry	2–3	dimitic, skeletal hyphae IKI–	fusiform	ellipsoid to short cylindrical, 8.5–11.5 × 4.5–6.5 μm	absent	[8,32]
*H. odorus*	China	perennial, effused-reflexed to pileate	white to cream when fresh, pale buff to light brown when dry	3–5	trimitic, skeletal hyphae IKI–	absent	ovoid, 5.5–6 × 4–5 μm	present	[5,8]
*H. papyraceus*	USA	annual, resupinate	white to cream when fresh, cream to buff when dry	3–4	dimitic, skeletal hyphae dextrinoid	fusiform	cylindrical, 13–15 × 5–6 μm	present	[31]
*H. pileatus*	Brazil	annual, pileate	ochraceous buff	3–4	dimitic, skeletal hyphae dextrinoid	absent	cylindrical, 9–10 × 4–5 µm	absent	[7]
*H. pirongia*	New Zealand	annual, resupinate	white to cream when fresh, pale brownish when bruised, pinkish buff to clay-buff when dry	3–4	trimitic, skeletal hyphae IKI–	fusiform	oblong ellipsoid to cylindrical, 11–14 × 5.2–7 µm	absent	[9]
*H. punctatus*	Sri Lanka	annual, resupinate	pale buff when dry	3–5	dimitic, skeletal hyphae dextrinoid	fusiform occasionally with a simple septum	cylindrical to slightly allantoid, 9–10.8 × 3.8–5 µm	absent	Present study
*H. srilankensis*	Sri Lanka	perennial, resupinate	pale buff when fresh, pinkish buff to salmon when dry	4–5	dimitic, skeletal hyphae dextrinoid	fusiform	oblong ellipsoid, 8.5–11 × 4–5.2 µm	present	Present study
*H. septatus*	China	annual, resupinate	white to cream when fresh, light buff when dry	5–6	dimitic, skeletal hyphae dextrinoid	fusiform	oblong to ellipsoid, 8.5–11 × 5–6 μm	absent	[5]
*H. subpapyraceus*	China	annual, resupinate	white to cream when fresh, cream to buff- yellow when dry	3–5	dimitic, skeletal hyphae dextrinoid	clavate with a simple septum	widely ellipsoid, 9–12 × 5.5–8 μm	absent	[5]
*H. subtrameteus*	Russia	perennial, resupinate	incarnadine when fresh, light reddish brown when dry	3–4	dimitic to trimitic, skeletal hyphae IKI–	fusiform	oblong ellipsoid to ellipsoid, 8.5–11 × 4.5–6.5 μm	absent	[5]
*H. thindii*	India	annual to perennial, resupinate	cream when fresh, clay-buff when dry	3–4	dimitic, skeletal hyphae IKI–	fusiform	cylindrical, 10.5–14.5 × 5.2–6.5 μm	absent	[5,8]
*H. tuberculosus*	Belarus	annual to perennial resupinate	cream when fresh, olivaceous buff when dry	2–3	trimitic, skeletal hyphae dextrinoid	fusiform	oblong ellipsoid, 13–15 × 6–8.5 μm	absent	[32]

## Data Availability

Publicly available datasets were analyzed in this study. This data can be found here: [https://www.ncbi.nlm.nih.gov/; https://www.mycobank.org/page/Simple%20names%20search; http://purl.org/phylo/treebase, submission ID 27556].

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
