# Peer review of "Molecular Phylogeny and Global Diversity of the Genus Haploporus (Polyporales, Basidiomycota)"

_jof, 2021, doi:10.3390/jof7020096_

Round 1

Reviewer 1 Report

Here is the review of manuscript entitled "Molecular phylogeny and global diversity of the genus Haploporus (Polyporales, Basidiomycota)".

The object of the paper was to update phylogeny and taxonomy of the genus Haploporus using a large number of samples from East & South Asia, Europe, and America. Additionally, four species are described as new to science on the basis of their morphological and molecular phylogenetic characters (ITS, nLSU, and mtSSU marker genes). All newly described species are compared to the most similar Haploporus species. Worldwide identification key to the genus Haploporus is presented as well.

The research methods used in the study are suitable and well conducted. The topic is very interesting for JoF audience. The English language used in the manuscript is pretty well. I am recommending the manuscript for publication after the implementation of minor corrections listed below.

A list of proposed corrections in the text:

4 Yu-cheng -> Yu-Cheng

18 (nLSU), the small -> (nLSU), and the small

18-19 bicolor -> bicolor

32 characterized as annual -> characterized as having annual

36 Kotlaba and Pouzar described -> Kotlaba and Pouzar [2] described

37 Dai et al. [2, 6]. -> Dai et al. [6].

49 illustrate these new -> illustrate those new

55 will forward to -> will forward specimens to

55 National Museum Prague of Czech Republic -> National Museum of Czech Republic in Prague

56 Sections were studied -> Sections of basidiocarps were studied

60 Spores -> Basidiospores

65 spores -> basidiospores

66 all spores -> all basidiospores

66 spores measured -> basidiospores measured

88 followed -> following

89 as outgroups -> as an outgroup

156 Among them two -> Among them are two

159 punctatus -> punctatus

170 Type (comment: Please include geographical coordinates of holotype collection)

171 (comment: who collected the holotype specimen? Please include name of the collector.)

176 at center -> at the center

202 no-dextrinoid -> non-dextrinoid

205 Type (Please include geographical coordinates of holotype collection)

206 (comment: who collected the holotype specimen? Please include name of the collector.)

210 at center -> at the center

242 Type (Please include geographical coordinates of holotype collection)

243 (comment: who collected the holotype specimen? Please include name of the collector.)

285 Type (Please include geographical coordinates of holotype collection)

286 (comment: who collected the holotype specimen? Please include name of the collector.)

289-290 at center -> at the center

303 ampullaceal -> ampullaceous

316 c (delete)

317 of species in Haploporus. -> of Haploporus species.

323 same size -> same-sized

332 sharing -> share

341 Figuresc 1–2 -> Figures 1–2

348 which > 2% -> which amounts to > 2%

356 closer to -> close to

357 discussed as above -> discussed above

361 basidiocarp -> basidiocarps

415 performe -> performing

432 Pachykytopsora -> Pachykytospora

432 Èeská -> Česká

Best,

Reviewer

Author Response

Dear reviewer,

We are very grateful to you for your patient modifications and comments on our manuscript. We have carefully revised the manuscript with the "Track Changes" function in Microsoft Word according to all the comments. The responses to the comments were listed below in red.

Point 1: 4 Yu-cheng -> Yu-Cheng

Response 1: It is done.

Point 2: 18 (nLSU), the small -> (nLSU), and the small

Response 2: It is done.

Point 3: 18-19 bicolor -> bicolor

Response 3: It is done.

Point 4: 32 characterized as annual -> characterized as having annual

Response 4: It is done.

Point 5: Kotlaba and Pouzar described -> Kotlaba and Pouzar [2] described

Response 5: It is done.

Point 6: 37 Dai et al. [2, 6]. -> Dai et al. [6].

Response 6: It is done.

Point 7: 49 illustrate these new -> illustrate those new

Response 7: It is done.

Point 8: 55 will forward to -> will forward specimens to

Response 8: It is done.

Point 9: 55 National Museum Prague of Czech Republic -> National Museum of Czech Republic in Prague

Response 9: It is done.

Point 10: 56 Sections were studied -> Sections of basidiocarps were studied

Response 10: It is done.

Point 11: 60 Spores -> Basidiospores

Response 11: It is done.

Point 12: 65 spores -> basidiospores

Response 12: It is done.

Point 13: 66 all spores -> all basidiospores

Response 13: It is done.

Point 14: 66 spores measured -> basidiospores measured

Response 14: It is done.

Point 15: 88 followed -> following

Response 15: It is done.

Point 16: 89 as outgroups -> as an outgroup

Response 16: We prefer to outgroups because we have two species as outgroups.

Point 17: 156 Among them two -> Among them are two

Response 17: It is done.

Point 18: 159 punctatus -> punctatus

Response 18: It is done.

Point 19: 170 Type (comment: Please include geographical coordinates of holotype collection)

Response 19: It is done.

Point 20: 171 (comment: who collected the holotype specimen? Please include name of the collector.)

Response 20: It is done.

Point 21: 176 at center -> at the center

Response 21: It is done.

Point 22: 202 no-dextrinoid -> non-dextrinoid

Response 22: It is done.

Point 23: 205 Type (Please include geographical coordinates of holotype collection)

Response 23: It is done.

Point 24: 206 (comment: who collected the holotype specimen? Please include name of the collector.)

Response 24: It is done.

Point 25: 210 at center -> at the center

Response 25: It is done.

Point 26: 242 Type (Please include geographical coordinates of holotype collection)

Response 26: It is done.

Point 27: 243 (comment: who collected the holotype specimen? Please include name of the collector.)

Response 27: It is done.

Point 28: 285 Type (Please include geographical coordinates of holotype collection)

Response 28: It is done.

Point 29: 286 (comment: who collected the holotype specimen? Please include name of the collector.)

Response 29: It is done.

Point 30: 289-290 at center -> at the center

Response 30: It is done.

Point 31: 303 ampullaceal -> ampullaceous

Response 31: It is done. Point 32: 316 c (delete) Response 32: It is done.

Point 33: 317 of species in Haploporus. -> of Haploporus species.

Response 33: It is done.

Point 34: 323 same size -> same-sized

Response 34: It is done.

Point 35: 332 sharing -> share

Response 35: It is done.

Point 36: 341 Figuresc 1–2 -> Figures 1–2

Response 36: It is done.

Point 37: 348 which > 2% -> which amounts to > 2%

Response 37: It is done.

Point 38: 356 closer to -> close to

Response 38: It is done.

Point 39: 357 discussed as above -> discussed above

Response 39: It is done.

Point 40: 361 basidiocarp -> basidiocarps

Response 40: It is done.

Point 41: 415 performe -> performing

Response 41: It is done.

Point 42: 432 Pachykytopsora -> Pachykytospora

Response 42: It is done.

Point 43: 432 Èeská -> Česká

Response 43: It is done.

Warm regards,

Meng Zhou, Yu-Cheng Dai, Josef Vlasák and Yuan Yuan

Reviewer 2 Report

The paper is ok.

I made only very few remarks shown in the attached manuscript version.

Author Response

Dear reviewer,

We are very grateful to you for your patient modifications and comments on our manuscript. We have carefully revised the manuscript with the "Track Changes" function in Microsoft Word according to all the comments. The responses to the comments were listed below in red.

Point 1: 13 taxonomy of Haploporus > taxonomy of the genus Haploporus

Response 1: It is done.

Point 2: 36 Pol-yporus > Poly-porus

Response 2: Due to format limitation, there is a word branch, and I have made corresponding modifications

Point 3: 83 Have you not done a gel electrophoresis before giving samples to sequencing?

Response 3: The part of job was handed by sequencing companies (see the revised manuscript).

Point 4: 115 what means j?

Response 4: jModelTest is the name of a tool to carry out statistical selection of best-fit models of nucleotide substitution.

Point 5: 316 remove this c

Response 5: It is done.

Point 6: 317 place this on top of the Table

Response 6: It is done.

Point 7: Table1 Basidiomata > Basidioma-ta; Dendrohyphidia > Dendroh-yphidia

Response 7: Due to the table format limitation, it is OK now.

Point 8: 342, 347, 349, 353, Haploporus > H.

Response 8: We prefer to the full genus name when it firstly appeared in each sentence, and the same genus is abbreviated in this sentence.

Warm regards,

Meng Zhou, Yu-Cheng Dai, Josef Vlasák and Yuan Yuan